# Association between acculturation and physician trust for internal migrants: A cross-sectional study in China

Enhong Dong[1,2,3☯], Ting Xu[1,2,3☯], Xiaoting Sun[4☯], Tao Wang[5,6], Yang Wang[7]*, Jiahua Shi[8]*

1 School of Nursing and Health Management, Shanghai University of Medicine & Health Science, Pudong New District, Shanghai, China, 2 School of Media and Communication, Shanghai Jiao Tong University, Shanghai, China, 3 Institute of Healthy Yangtze River Delta, Shanghai Jiao Tong University, Shanghai, Shanghai, China, 4 Shanghai Tongji Hospital, Shanghai, China, 5 Department of Emergency, Shanghai East Hospital, Tongji University School of Medicine, Shanghai, China, 6 College of Arts and Media, Tongji University, Shanghai, China, 7 China Center for Health Development Studies, Peking University, Beijing, China, 8 HuangPu District Health Promotion Center, ShangHai, China

☯ These authors contributed equally to this work.
* shijiahuaabc@163.com (JS); yang.wang@hsc.pku.edu.cn (YW)

**Data Availability Statement:** All relevant data are within the paper and its Supporting Information files.

## Abstract

### Background

Physician trust is a critical determinant of the physician–patient relationship and is necessary for an effective health system. Few studies have investigated the association between acculturation and physician trust. Thus, this study analyzed the association between acculturation and physician trust among internal migrants in China by using a cross-sectional research design.

### Methods

Of the 2000 adult migrants selected using systematic sampling, 1330 participants were eligible. Among the eligible participants, 45.71% were female, and the mean age was 28.50 years old (standard deviation = 9.03). Multiple logistic regression was employed.

### Results

Our findings indicated that acculturation was significantly associated with physician trust among migrants. The length of stay (LOS), the ability of speaking Shanghainese, and the integration into daily life were identified as contributing factors for physician trust when controlling for all the covariates in the model.

### Conclusion

We suggest that specific LOS-based targeted policies and culturally sensitive interventions can promote acculturation among Shanghai's migrants and improve their physician trust.

**Funding:** –E.D. T.W. –Grant No. 19BGL246 and Grant No. 21692104900:E.D.;Grant No. 18ZDA088: T.W. –National social Science Foundation of China General Project (Grant No. 19BGL246); National social Science Foundation of China Major Project (Grant No. 18ZDA088); Shanghai 2021 "Science and Technology Innovation Action Plan" Soft Science Key Project (Grant No. 21692104900) –http://www.nopss.gov.cn/GB/index.html http://stcsm.sh.gov.cn/ –The funders had no role in study design, data collection and analysis, decision to publish, or preparation of the manuscript.

**Competing interests:** The authors have declared that no competing interests exist.

# Background

Physician trust refers to a patient's optimistic acceptance of a vulnerable medical situation and belief that their physician is willing to care for the patient's health and interests [1, 2]. Studies have revealed that physician trust is a strong predictor of health outcomes, such as patient satisfaction, health-seeking behaviors, continuity of care, and adherence to treatment [3, 4]. Physician trust facilitates patients' access to health care and their disclosure of relevant information, which enables accurate and timely diagnoses [2], improves the self-reported health status of patients, and enhances the patient's ability to manage chronic diseases [5]. Due to the information asymmetry between physicians and patients, patients' trust in their physician is the basis of the physician–patient relationship [6]. The patient's trust in their physician is the patient's expectation that their physician will provide beneficial care and truthful information, regardless of the patient's ability to monitor the physician [7]. Trust building is a key step in developing high-quality physician–patient interactions and relationships [8]. Distrustful patients are suspicious of their physician's motivations and may behave aggressively. Many studies have identified that physician trust is a critical element of the physician–patient relationship and is necessary for building an effective health-care system [1, 4, 9]. Achieving high levels of physician trust and identifying sources of mistrust are key goals for health-care policy [4].

Acculturation is the process of cultural change and adaption or maladaptation that stems from contact with culturally different people, groups, and social influences [10]. Some scholars have applied immigration acculturation theory to internal migrants in China [11–13] because China hosts a large migratory population with massive intracultural differences between rural and urban regions [14, 15]. These differences are often demonstrated in terms of behaviors, psychology, interpersonal communication (e.g., language use) [16], and other social and cultural differences (e.g. food, clothing, customs, and social interactions). With the rapid increase in urbanization in China, many internal migrants who lived in a place other than their officially registered residence (*hukou*) for at least 6 months have gradually moved to urban areas in search of better educational and career opportunities. Despite many internal migrants' strong desire to stay and become permanent residents in new cities, the adaptation and acculturation process is often long and stressful [12]. Due to economic and cultural discrepancies between urban and rural regions, migrants often experience social exclusion and prejudice when adjusting to city life. Furthermore, long-distance migration and structural barriers (e.g., *hukou*, health insurance, and perceived discrimination) can greatly increase the likelihood of cultural distance, which subsequently increases the chance of difficulties related to language, values, social networks, culture, and lifestyle [17, 18]. According to a report from the *Xinhua News Agency* in 2018, 260 million outpatient and emergency department visits and 4.4733 million discharged patients were recorded in Shanghai medical institutions, of which internal migrants accounted for approximately 30% [19]. Although structural barriers have widened the existing gap between the restricted supply of and increased demand for health services, relationships between physicians and migrant patients have deteriorated in Shanghai; internal migrants were involved in approximately 60% of violent events against physicians and physician–patient dispute cases in Shanghai in 2017 [20]. Many studies have supported that acculturation is a significant predictor of health service encounters, including physician trust and health care [21–24]. Therefore, to address the association of acculturation with physician trust, this article examines the association of acculturation with physician trust among migrant Shanghainese patients and the contributing acculturative factors.

To identify the association between acculturation and physician trust among internal migrants, other variables must be controlled for. Therefore, we accounted for age; sex; educational level; type of insurance; and health profile, including outpatient visit frequency, number

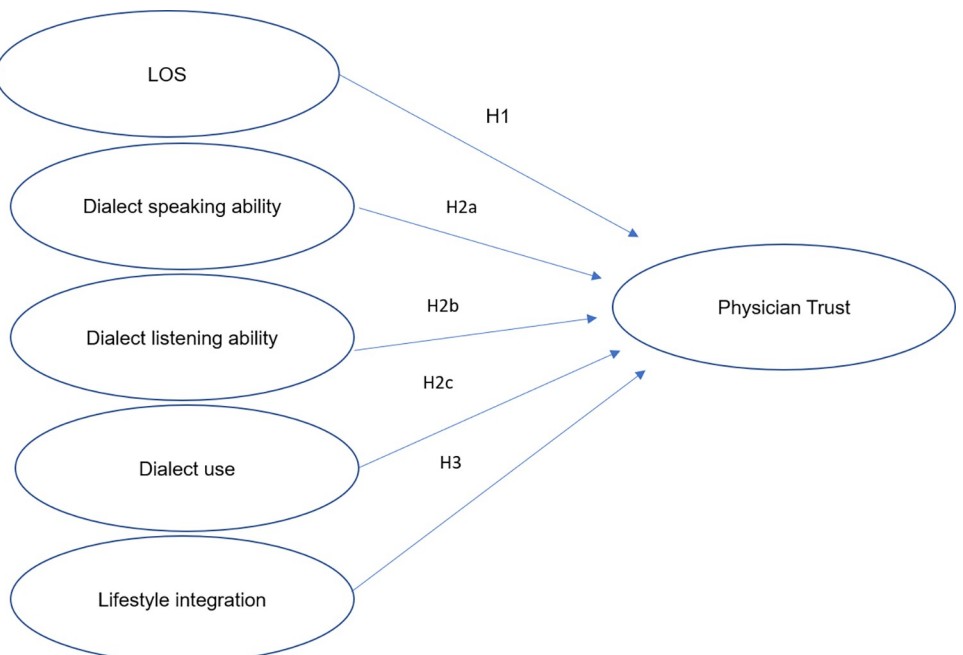

**Fig 1. Conceptual model based on immigration acculturation theory.**

of chronic diseases, and self-reported health status [25–27], to identify the significance of the association between acculturation and physician trust.

We constructed a conceptual framework based on immigration acculturation theory (Fig 1). Acculturation is often conceptualized as a process through which an immigrant adopts the language, customs, food, behaviors, and attitudes of the host culture [28]. Acculturation has been linked to physical and psychological outcomes, such as obesity [29], morbidity [30], mortality [31], satisfaction with urban life [32], psychological intentions (e.g., suicide intention) [33], and physician trust [34]. Although acculturation is positively associated with some physical health disorders, psychiatric disorders, and healthy behaviors among some populations [35], patient satisfaction and physician trust have also increased after acculturation among other populations [34, 36]. Many studies have indicated that immigrants' reported trust in health-care providers increases with greater acculturation following immigration [25, 37, 38]. In other words, as migrants adjusted to their host environment, they reported better experiences during health visits and became more positive toward the quality of health care. Due to the uncertain nature of these findings, they have seldom been applied to preventive or intervention programs in physician–patient relationship management. Most studies examining the association between acculturation and physician trust among ethnocultural groups have examined Latin American and Asian patients; however, few studies have focused on internal Chinese migrants.

By using the conceptual model, we hypothesized that physician trust tends to increase over time among internal migrants as LOS gradually increases (H1). Additionally, we expected that language proficiency is positively associated with physician trust. We separated H1 into three subhypotheses: Shanghainese speaking proficiency is positively associated with physician trust (H2a); Shanghainese listening ability is positively associated with physician trust (H2b); and Shanghainese use at home, work, and with friends is positively associated with physician trust (H2c). Finally, we hypothesized that lifestyle integration is positively associated with physician trust (H3).

## Materials and methods

### Participants

The survey for this study was conducted at the Huangpu Physical Examination Center of Shanghai from June to September 2019. We used a systematic sampling method to select eligible participants. Participants were included in the study if they (1) were 16 years or older, (2) were not originally born in Shanghai but have been granted legal permanent or temporary residency permits by the Migrant Population Management Office for at least 6 months, and (3) visited a physician in Shanghai more than once in the past year. This study was conducted in compliance with the Shanghai University of Medicine and Health Sciences Institutional Review Board for the Protection of Human Subjects on March 5, 2019 (No:2019-gskyb-02-372424198012222511). All participants provided written informed consent.

### Measurements

The data were collected using a self-developed questionnaire containing 27 questions regarding participants' socio-demographics, health profiles (health-care use and health status), acculturation, and physician trust (see S1 Table and S1 File).

Studies have measured acculturation by using a standardized scale, such as a Marin short acculturation scale [39]; an Acculturation Rating Scale for Mexican Americans-II (ARSMA-II); or proxy measures, including LOS- and language-based questions, to assess the level of acculturation. In particular, proxy measures have demonstrated a high correlation with other more frequently used standardized scales [40, 41]. Therefore, in this study, we measured acculturation through four proxies: LOS in Shanghai, dialect proficiency, dialect use, and lifestyle integration. LOS in Shanghai was self-reported and categorized into the following four groups: less than 1 year (referent), 1–5 years, 5–10 years, and 10 or more years. Dialect proficiency was measured using two items that required participants to rate their dialect speaking and listening abilities on a 4-point Likert scale with the endpoints of 1 (very well) and 4 (not at all). Dialect use was measured using three items regarding the choice of dialect during different occasions (i.e., at home, at work, and with friends). Lifestyle integration was measured using a Chinese version of the acculturation scale [42], which was revised using the existing acculturation scales developed by other scholars [43–45]. The scale included four questions regarding dietary habits, dressing, entertainment, and social customs. Each question was related to the degree of changes since moving to Shanghai and was rated on a 5-point Likert scale, with the endpoints of 1 (*completely the same*) and 5 (*completely different*). A score closer to 1 indicated little-to-no lifestyle integration or a lower acceptance of the host culture, and a score closer to 5 indicated high lifestyle integration or a higher acceptance of the host culture. Because life integration was measured ordinally, it was considered a binary variable (i.e., high vs. low) in this study for statistical simplicity. The cut-off point was set at a sample median used in another study [46]. Reliability was examined using Cronbach's $\alpha$. Cronbach's $\alpha$ for the four items related to lifestyle integration was 0.870.

Physician trust was the dependent variable in this study and was measured using a 11-item Chinese version scale for physician trust in patients, which was revised for evaluating Chinese patients and medical contexts [47]. The scale has been used in many studies to measure physician trust in the Chinese health-care context and has demonstrated good reliability and validity [48, 49]. In this study sample, Cronbach's $\alpha$ for physician trust was 0.91.

### Statistical analysis

Participant demographics and socioeconomic characteristics were summarized using descriptive analyses. Bivariate analyses were conducted to examine the association between

acculturation variables and physician trust scores by performing a *t* test for continuous variables and a Chi-square test for categorical variables. Multivariate logistic regression models were performed to calculate the effects of acculturation on physician trust among migrants. Predictors that were significant at a liberal *p* value of <0.05 were retained as candidates for the multivariable model. The covariates in the regression models included age, sex, education level, type of insurance, outpatient visit frequency, and other confounders. We constructed and compared three models to identify a significant association between acculturation variables and physician trust. We used a directed acyclic graph to illustrate the relationship between acculturation and physician trust, including previously mentioned covariates. The covariates that we controlled for were chosen on the basis of findings in the literature [25–27] (S1 Fig). Three models were constructed using the manual forward selection procedure; sociodemographic variables (i.e., sex and insurance type) were significant in the bivariate analysis, and the health profile variables (i.e., outpatient visit frequency, number of chronic diseases, and self-reported health status) were sequentially added to the model. Specifically, Model 1 assessed the unadjusted association between acculturation variables and physician trust, Model 2 adjusted for the effect of sociodemographic variables, and Model 3 adjusted for both sociodemographic and health profile variables. Age and educational level were key confounders, which were also adjusted in the model irrespective of their *p* values.

All analyses were conducted using Stata.15.0 (Stata, College Station, TX, USA).

## Results

### Descriptive analysis of participants

Among 2000 participants, 1480 were eligible participants, 1390 participants (response rate of 93.92%) returned the questionnaires, and 1330 were included in the final analyses after excluding those with >5% missing data. The final sample included 1330 participants (46% women, $M_{age}$ = 28.50 years, standard deviation = 9.03) who met the inclusion criteria. Of the 1330 participants, 1178 (88.57%) reported a high level of physician trust in Shanghai. Additional characteristics of participants in analysis stratified by the level of physician trust are displayed in Table 1.

### Bivariate analysis

Significant covariates identified in bivariate analyses were female sex, insurance type, and a high frequency of visiting physicians. A greater number of chronic diseases and worse self-reported health were associated with the less likelihood of reporting a high level of physician trust. However, a good health status was associated with a higher likelihood of reporting a high level of physician trust. No significant difference was observed in physician trust in analyses based on educational level, marital status, employment type, income level, physical examinatioFn frequency, number major diseases, district of residency. Significant acculturation variables associated with physician trust in bivariate analyses were LOS years in Shanghai, Shanghainese dialect speaking and listening, and lifestyle integration. Additional sociodemographic factors based on the level of physician trust are presented in Table 1.

### Logistic analysis

In Model 1, we identified a positive association between LOS in the host region and physician trust (Table 2). Compared with the participants who lived in Shanghai for less than 1 year, those who lived in Shanghai for 1–5 years were 1.36 times more likely to have a higher level of

**Table 1. Sample characteristics and univariate analysis of physician trust in internal migrants in Shanghai (n = 1330).**

| Variables | n(%) | Physician Trust | | P value |
|---|---|---|---|---|
| | | High level n(%) | Low level n(%) | |
| **Physician Trust(n = 1330)** | | **1178** | **152** | |
| Age | 28.50±9.03 | -- | -- | 0.093 |
| Sex(n = 1330) | | | | |
| Female | 608(45.71) | 524(39.40) | 84(6.32) | 0.012 |
| Male | 722(54.29) | 654(49.17) | 68(5.11) | |
| Educated level(n = 1330) | | | | |
| Lower or equal to primary education | 35(2.63) | 30(2.26) | 5(0.38) | 0.200 |
| Secondary education | 984(73.98) | 882(66.32) | 102(7.67) | |
| College education | 309(23.23) | 264(19.85) | 45(3.38) | |
| Higher or equal to graduated education | 2(0.15) | 2(0.15) | 0(0.00) | |
| Marital type(n = 1330) | | | | |
| Never married | 767(57.67) | 690(51.68) | 77(5.79) | 0.065 |
| Married | 563(42.33) | 488(36.69) | 75(5.64) | |
| Employed type(n = 1330) | | | | |
| Unemployed | 41(3.08) | 40(3.01) | 1(0.07) | 0.185 |
| Employed | 1169(87.89) | 1032(77.59) | 137(10.30) | |
| Self-employed | 120(9.02) | 106(7.97) | 14(1.05) | |
| Retired | 0(0.00) | 0(0.00) | 0(0.00) | |
| Insured type(n = 1329) | | | | |
| Uninsured | 242(18.21) | 225(16.93) | 17(1.28) | 0.040 |
| NRCMI | 407(30.62) | 352(26.49) | 55(4.14) | |
| UEBMI/URBMI | 680(51.17) | 600(45.15) | 80(6.02) | |
| Annual income level(n = 1330) | | | | |
| <50,000 RMB | 429(32.26) | 382(28.72) | 47(3.53) | 0.969 |
| 50,000–100,000 RMB | 596(44.81) | 528(39.70) | 68(5.11) | |
| 100,000–250,000 RMB | 246(18.50) | 216(16.24) | 30(2.26) | |
| > = 250,000 RMB | 59(4.44) | 52(3.91) | 7(0.53) | |
| Annual physical examination frequency (n = 1330) | | | | |
| 0 time | 131(9.85) | 113(8.50) | 18(1.35) | 0.685 |
| 1 time | 958(72.03) | 851(63.98) | 107(8.05) | |
| 2 times | 209(15.71) | 187(14.06) | 22(1.65) | |
| > = 3 times | 32(2.41) | 27(2.03) | 5(0.38) | |
| Outpatient Visit frequency(n = 1330) | | | | 0.001 |
| 1 time | 550(41.35) | 506(38.05) | 44(3.31) | |
| 2 times | 317(23.83) | 281(21.13) | 36(2.71) | |
| > = 3 times | 463(34.81) | 391(29.40) | 72(5.41) | |
| District(n = 1330) | | | | 0.098 |
| Metropolitan area | 998(75.04) | 894(67.22) | 104(7.82) | |
| Fringe areas | 209(15.71) | 181(13.61) | 28(2.11) | |
| Suburban areas | 123(9.25) | 103(7.74) | 20(1.50) | |
| Number of chronic diseases (n = 1328) | | | | <0.001 |
| 0 | 1069(80.50) | 961(72.36) | 108(8.13) | |
| 1 | 227(17.09) | 193(14.53) | 34(2.56) | |
| > = 2 | 32(2.41) | 22(1.66) | 10(0.75) | |
| Number of major diseases (n = 1327) | | | | 0.925 |
| 0 | 1319(99.40) | 1168(88.02) | 151(11.38) | |
| 1 | 8(0.60) | 7(0.53) | 1(0.08) | |

*(Continued)*

**Table 1.** (Continued)

| Variables | n(%) | Physician Trust | | P value |
|---|---|---|---|---|
| | | High level n(%) | Low level n(%) | |
| **Physician Trust(n = 1330)** | | **1178** | **152** | |
| > = 2 | 0(0.00) | 0(0.00) | 0(0.00) | |
| **Health status (n = 1329)** | | | | |
| High level | 1201(90.37) | 1088(81.87) | 113(8.50) | <0.001 |
| Low level | 128(9.63) | 89(6.70) | 39(2.93) | |
| **Length of years in Shanghai (n = 1324)** | | | | |
| <1 year | 267(20.17) | 241(18.20) | 26(1.96) | 0.044 |
| 1–5 years | 492(37.16) | 447(33.76) | 45(3.40) | |
| 5–10 years | 236(17.82) | 201(15.18) | 35(2.64) | |
| > = 10 years | 329(24.85) | 284(21.45) | 45(3.40) | |
| **Shanghainese speaking ability (n = 1329)** | | | | 0.048 |
| Not well | 296(22.27) | 265(19.94) | 31(2.33) | |
| well | 1033(77.73) | 912(68.62) | 121(9.10) | |
| **Shanghainese listening ability (n = 1329)** | | | | 0.045 |
| Not well | 658(49.51) | 591(44.47) | 67(5.04) | |
| well | 671(50.49) | 586(44.09) | 85(6.40) | |
| **Dialect use at home(n = 1330)** | | | | 0.829 |
| Shanghai dialect | 40(3.01) | 35(2.63) | 5(0.38) | |
| Not Shanghai dialect | 1290(96.99) | 1143(85.94) | 147(11.05) | |
| **Dialect use at work(n = 1330)** | | | | 0.756 |
| Shanghai dialect | 55(4.14) | 48(3.61) | 7(0.53) | |
| Not Shanghai dialect | 1275(95.86) | 1130(84.96) | 145(10.90) | |
| **Dialect use with friends(n = 1330)** | | | | |
| Shanghai dialect | 43(3.23) | 39(2.93) | 4(0.30) | 0.656 |
| Not Shanghai dialect | 1287(96.77) | 1139(85.64) | 148(11.13) | |
| **Lifestyle integration** | | | | 0.017 |
| Low | 158(11.90) | 147(11.07) | 11(0.83) | |
| High | 610(45.93) | 537(40.44) | 73(5.50) | |

NRCMI: the New Rural Cooperative Medical Insurance; UEBMI: Urban Employee Basic Medical Insurance;URBMI: Urban Resident Basic Medical Insurance

physician trust (odds ratio [OR]: 1.36, 95% confidence interval [CI]: 1.04–1.78). Moreover, those who lived in Shanghai for 5–10 years and 10 or more years were 1.78 and 2.70 times more likely to have a higher level of physician trust, respectively (OR: 1.78, 95% CI: 1.66–3.97; OR: 2.70, 95% CI: 2.58–4.93, respectively) compared with those who lived in Shanghai for less than 1 year. Moreover, those who reported high speaking and listening proficiency in Shanghainese were more likely to report higher physician trust than those who did not (OR: 1.56, 95% CI: 1.35–2.04; OR: 1.74, 95% CI: 1.50–2.11, respectively). Additionally, those who had a higher degree of lifestyle integration were more likely to report a higher level of physician trust compared with those who reported a lower degree of lifestyle integration (OR: 1.65, 95% CI: 1.30–2.11). However, using Shanghainese at home, work, and with friends was not significantly associated with physician trust in Model 1.

In Model 2, we identified similar results as those in Model 1. First, compared with those who lived in Shanghai for less than 1 years, participants who lived in Shanghai for 1–5 years were 1.25 times more likely to report a higher level of physician trust (OR: 1.25, 95% CI: 1.02–

**Table 2. Multivariable logistic regression models to assess the association between immigrant's acculturation and physician trust in internal migrants in Shanghai (n = 1330).**

| Acculturation variables | Model 1 OR[a] Unadjusted[#] | Model 2 OR[b] Adjusted for demographic and socio-economic covariates[∫] | Model 3 OR[c] Adjusted for additional health profile based on model 2[†] |
|---|---|---|---|
| **Length of years in Shanghai (n = 1324)** | | | |
| <1 year | 1.00 | 1.00 | 1.00 |
| 1–5 years | 1.36(1.04–1.78) ** | 1.25(1.02–1.59) ** | 1.23(1.01–1.49) ** |
| 5–10 years | 1.78(0.66–0.97) ** | 1.69(0.58–0.87) ** | 1.61(0.48–0.93) ** |
| > = 10 years | 2.70(0.58–0.93)** | 2.68(0.50–0.80) ** | 2.59(0.33–0.84) ** |
| **Shanghainese speaking ability (n = 1329)** | | | |
| Not well | 1.00 | 1.00 | 1.00 |
| Well | 1.56(1.35–2.04) ** | 1.55(1.33–2.06) ** | 1.37(1.31–2.11) * |
| **Shanghainese listening ability (n = 1329)** | | | |
| Not well | 1.00 | 1.00 | 1.00 |
| Well | 1.74(1.50–2.11)** | 1.78(1.52–2.17)** | 1.68(0.82–2.47) |
| **Dialect use at home(n = 1330)** | | | |
| Not Shanghai dialect | 1.00 | 1.00 | 1.00 |
| Shanghai dialect | 0.65(0.20–2.13) | 0.76(0.24–2.45) | 0.71(0.22–2.32) |
| **Dialect use at work(n = 1330)** | | | |
| Not Shanghai dialect | 1.00 | 1.00 | 1.00 |
| Shanghai dialect | 0.83(0.32–2.15) | 0.79(0.31–2.00) | 0.74(0.28–1.96) |
| **Dialect use with friends(n = 1330)** | | | |
| Not Shanghai dialect | 1.00 | 1.00 | 1.00 |
| Shanghai dialect | 1.91(0.67–5.32) | 1.79(0.55–5.85) | 1.68(0.51–5.57) |
| **Lifestyle integration** | | | |
| low | 1.00 | 1.00 | 1.00 |
| high | 1.65(1.30–2.11)** | 1.62(1.26–2.17)** | 1.56(1.20–2.27)* |

[a] Unadjusted crude model

[b] Multivariate model adjusted for age, sex, educational attainment, insured type

[c] Multivariate model adjusted for outpatient visit frequency, number of chronic diseases, and self-reported health status as well as socio-demographics

*p<0.05

** p<0.001

1.59), and those who lived in Shanghai for 5–10 years and 10 or more years were 1.69 and 2.68 times more likely to report a higher level of physician trust, respectively (OR: 1.69, 95% CI: 1.58–3.87; OR: 2.68, 95% CI: 2.50–4.80, respectively).

Second, participants who reported high listening and speaking proficiency in Shanghainese also had a higher level of physician trust compared with their counterparts (OR: 1.55, 95% CI: 1.33–2.06; OR: 1.78, 95% CI: 1.52–2.17, respectively). Furthermore, those who had a higher degree of lifestyle integration were more likely to report a higher level of physician trust (OR: 1.62, 95% CI: 1.26–2.17) compared with those who had a lower degree of lifestyle integration.

However, using Shanghainese at home, work, and with friends was not significantly associated with physician trust in Model 2.

In Model 3, when controlling for all the variables, we found that only LOS in Shanghai, high speaking proficiency of Shanghainese, and lifestyle integration were significantly associated with a high level of physician trust. However, the listening ability of Shanghainese was not a significant predictor.

## Discussion and conclusion

### Discussion

First, we found that LOS in Shanghai was associated with physician trust among Shanghai migrants, supporting H1. This finding is concordant with those of previous studies that LOS in host countries or regions was significantly associated with strong subjective assessment scores of health-care services among migrants [50, 51]. The most plausible explanation is that the duration of acculturation increased the reported health-related advantages among migrants, including physician trust [52]. For internal migrants, a migrant who has stayed longer in Shanghai may be more acculturated, experienced with navigating through the health care system, and able to communicate with health-care providers, all of which gradually improve trust in the health-care system and in physicians [34]. Accordingly, this effect may have caused the upward trend in physician trust among longer-term migrants in Shanghai.

Second, Shanghainese proficiency and use were an effective measurement of acculturation, although they might not have indicated assimilation to unhealthy norms or lifestyles. For some migrants, higher Shanghainese proficiency and use may improve integration and acceptance into host culture [53, 54]. Some studies have argued that poor language proficiency and less frequent use may lead to discrimination against migrants and their exclusion from some aspects of the host society [54, 55]. However, our study suggested that, when considering all covariates, only Shanghainese speaking ability positively affected physician trust. Thus, only H2a was supported. Speaking proficiency rather than listening ability could establish more patient–physician language concordance and subsequently facilitate the assimilation of migrants into Shanghai's local culture. For migrants in Shanghai, the physician–patient relationship was strengthened when migrants with a high proficiency of Shanghainese perceived themselves similar to their physicians in terms of beliefs, values, and language [56, 57]. Some studies have reported that perceived personal similarity is associated with higher levels of trust, satisfaction, and adherence [56–58]. Therefore, language proficiency may facilitate improvements to physician–patient relationships.

Third, some lifestyle changes manifested in attitudes, beliefs, values, and behaviors, such as dietary habits, clothes, entertainment, and social customs. Acculturation is "the process by which an ethnic group, usually a minority, adopts the cultural patterns, including beliefs, religion, and language, of a dominant group" [32] and has been conceived as a dynamic process in which individuals gradually adjust to a new environment. For internal migrants in Shanghai, having a higher degree of lifestyle integration indicated that migrants were more likely to adapt to the new host environment and accept the host culture through cultural and behavioral assimilation. Lifestyle is an essential factor for acculturation among internal migrants [59]. Although adopting the mainstream lifestyle may negatively affect mental health by evoking stress [60, 61], lifestyle adjustments can promote cultural communication with local physicians and subsequently improve psychological adjustment [32]. Lifestyle adjustments also affected physician trust among migrants in Shanghai; lifestyle integration positively affected migrants' medical visits, perceived quality of health-care services, and their relationship with physicians. Therefore, lifestyle integration was positively associated with physician trust, which supports

H3. This finding is consistent with the results of other studies [38, 62], which indicated that cultural and behavioral assimilation can influence physician trust among internal migrants.

This study has some limitations. First, this sample included only migrants who were granted legal permanent or temporary residency permits for 6 months or longer. Thus, migrants who did not obtain a legal permanent or temporary residency permit or had residency permits for less than 6 years were excluded. Moreover, participants were selected from one physical examination center, which prevented the generalizability of our findings. Further research has already been planned to evaluate patients from multiple centers. Second, the acculturation variables measured in this study may not have captured every aspect of acculturation. Acculturation is difficult to measure and has many aspects, such as values, interethnic interactions, cultural domains, participation, and identity, some of which are unobservable. Changes in acculturation can indicate how the original culture and new culture interact to produce new values, attitudes, and beliefs. In this study, we evaluated LOS, Shanghainese proficiency and use, and lifestyle integration to measure acculturation, which may have produced measurement bias. Therefore, using an alternative scale to measure the changes in psychological and behavioral acculturation, such as the Psychological–Behavioral Acculturation Scale [63], can ensure more robust results. Third, according to a study by Sakamoto [64], acculturation theory has some deficiencies. For example, acculturation theory mainly neglects structural issues that affect migrants experiencing unfamiliar cultures and may lead migrants to not being perceived as "sufficiently" assimilated. This can occur even if migrants experienced social exclusion or discrimination. Therefore, structural barriers, such as cultural distance [65], perceived discrimination [24], and *hukou* [66], should also be considered in future research. Fourth, the self-reported physician trust may have produced recall bias. For self-reported outcomes such as health-related quality of life, satisfaction and trust can vary depending on the event being recalled, time since the event, and the clinical and demographic characteristics of patients. The outcome measurement scores were always collected after patients sought health-care services, which may have caused recall bias that affected the actual trust scores. Therefore, physician trust should be supplemented with other measures to provide a broader understanding of the physician–patient relationship and health service quality.

## Contributions and policy implications

The present study has several contributions. First, this study can inform the Shanghai government of targeted measures for rebuilding physician trust among migrants and improving the quality of health-care services from an acculturative perspective. Policies can target temporary and permanent residency, employment, and education for migrants and their children to facilitate integration and assimilation. Because long-term internal migrants demonstrated a higher level of physician trust compared with those who lived in Shanghai for less than 1 year, policies should target long-term migrants to enhance their physician trust. Additionally, our findings inform strategies for developing community-based intervention programs that address the needs of migrants who have lived in Shanghai for less than 1 year. Such policies can help health-care policymakers and practitioners to develop culturally sensitive interventions for providing medical care that targets Shanghai's migrant community. Moreover, barriers between physicians and migrant patients can be broken down to improve physician trust and enhance the physician–patient relationship. Second, this study included several measures of acculturation. Our study presented a thorough examination of acculturation dynamics. By examining many aspects of acculturative progress in migration, we captured the specific acculturative factors that affect the migrants' physician trust and identified nonsignificant factors. Third, we examined a large sample of migrants in a metropolitan city in China. These migrants

routinely received physical examination in an authorized center that covered all regions of Shanghai, the second largest city in China. According to official statistics, approximately 15% of internal migrants are concentrated in the four mega-cities of Beijing, Shanghai, Guangzhou, and Shenzhen. Although geographical, economic, and cultural differences exist among the four cities, the internal migrants in Shanghai, which has one of the largest proportions of migrants, exhibit common characteristics with internal migrants in the other three cities. For example, internal migrants mainly belong to interprovincial flows; are likely to pursue higher wages, better employment positions, and better educational opportunities for their children; and must adapt to the local cultural and competitive environment [67]. Therefore, the findings of this study may be more generalizable to the national-level rather than to specific ethnic groups or geographic regions. Further research with a longitudinal study design that focuses on physician trust and the aspects of acculturation is required.

## Conclusion

This study established the association between acculturation and physician trust among internal migrants in Shanghai. LOS, Shanghainese speaking proficiency, and lifestyle integration were identified as contributing factors for physician trust when controlling for all acculturative variables. To improve physician trust among migrants in large cities in China, policies that target internal migrants based on their LOS and culturally sensitive interventions are required. Moreover, physician trust can be improved by fully considering the effects of acculturation in health-care settings, meeting the specific medical needs of internal migrants, and facilitating integration and assimilation into the host culture.

## Supporting information

**S1 Table. Information about the legal permanent system or temporary residency permit system in China.**
(DOCX)

**S1 File. The survey questionaire for the study.**
(DOCX)

**S1 Fig. DAG illustrating the relationship between acculturaion and physician trust, including possible covariates in the regression models.**
(JPG)

**S1 Dataset.**
(RAR)

## Acknowledgments

We thank all participants for their contribution in our study and the reviewers for the suggestions provided.

## Author Contributions

**Conceptualization:** Enhong Dong, Tao Wang, Jiahua Shi.

**Data curation:** Enhong Dong, Ting Xu, Xiaoting Sun, Yang Wang, Jiahua Shi.

**Formal analysis:** Yang Wang, Jiahua Shi.

**Funding acquisition:** Enhong Dong.

**Investigation:** Ting Xu, Xiaoting Sun, Jiahua Shi.

**Methodology:** Enhong Dong, Tao Wang.

**Supervision:** Tao Wang, Jiahua Shi.

**Writing – original draft:** Enhong Dong, Ting Xu, Yang Wang, Jiahua Shi.

**Writing – review & editing:** Enhong Dong, Xiaoting Sun, Yang Wang.

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
