## [Decision Letter · Decision Letter 0]

9 Aug 2022

PONE-D-22-15202Effects of acculturation on physician trust for internal migrants: a cross-sectional study in ChinaPLOS ONE

Dear Dr. shi,

Thank you for submitting your manuscript to PLOS ONE. After careful consideration, we feel that it has merit but does not fully meet PLOS ONE’s publication criteria as it currently stands. Therefore, we invite you to submit a revised version of the manuscript that addresses the points raised during the review process.

We look forward to receiving your revised manuscript.

Kind regards,

Wenhui Mao, PhD

Academic Editor

PLOS ONE

Journal Requirements:

2. PLOS ONE does not copy edit accepted manuscripts (https://journals.plos.org/plosone/s/criteria-for-publication#loc-5). To that effect, please ensure that your submission is free of typos and grammatical errors.

“Funding was provided by National social Science Foundation of China General Project (Grant No. 19BGL246); National social Science Foundation of China Major Project (Grant No. 18ZDA088); Shanghai 2021 "Science and Technology Innovation Action Plan" Soft Science Key Project (Grant No. 21692104900). The funders had no role in the question design, analysis or interpretation.”

 “--E.D.  T.W.

--Grant No. 19BGL246 and Grant No. 21692104900 :E.D.;Grant No. 18ZDA088:T.W.

--National social Science Foundation of China General Project (Grant No. 19BGL246); National social Science Foundation of China Major Project (Grant No. 18ZDA088); Shanghai 2021 "Science and Technology Innovation Action Plan" Soft Science Key Project (Grant No. 21692104900)

--http://www.nopss.gov.cn/GB/index.html

http://stcsm.sh.gov.cn/

--The funders had no role in study design, data collection and analysis, decision to publish, or preparation of the manuscript.”

Reviewers' comments:

Reviewer's Responses to Questions

**Comments to the Author**

1. Is the manuscript technically sound, and do the data support the conclusions?

Reviewer #1: Yes

Reviewer #2: Partly

2. Has the statistical analysis been performed appropriately and rigorously? 

Reviewer #1: Yes

Reviewer #2: Yes

3. Have the authors made all data underlying the findings in their manuscript fully available?

Reviewer #1: Yes

Reviewer #2: Yes

4. Is the manuscript presented in an intelligible fashion and written in standard English?

Reviewer #1: Yes

Reviewer #2: No

5. Review Comments to the Author

Reviewer #1: I think this is a policy-relevant research article. I thought that the definition of internal migrants may varied across time. The authors may give a clear objective study population under the circumstance of a cross-section data time.

Reviewer #2: Thank you for considering me as a reviewer for this manuscript in your esteemed journal. In general, I believe that it’s an interesting study. The authors focus on an important research question. It also has a great potential for developing a practical, culturally safe tool that can be used to improve the patient experience of migrant workers and promote social cohesion in China. This study may also have implications for other countries that are experiencing the rise of urbanization. However, there are some key issues that I suppose the authors need to address before the manuscript can be considered for publication:

Major concerns:

Introduction

Motivation

1. I suggest our authors re-emphasize the importance of physician trust in health service research as well as its relationship with other important psychosocial measures that they have mentioned (e.g., physician-patient relationships, ratings of providers). For example, is there any evidence to show that physician trust has an impact on the health service utilisation or the health outcomes of patients? Is there any overlap between physician-patient relationships and physician trust?

2. The authors mentioned that more than 60% of health care workplace violence in Shanghai was caused by migrant workers. It would have made the authors' research questions more meaningful if they had given the percentage of migrant workers in the population of Shanghai, or the percentage of migrant workers in outpatient/inpatient care utilisations.

Conceptual framework

3. The authors should explain why “immigration acculturation theory” is suitable for studying migrant workers in China (since it was designed for immigrants in the Western world) and how they developed their framework (based on existing literature? Expert opinions? Or something else?). From my point of view, many structural barriers (e.g., hukou, health insurance, perceived discrimination) faced by migrant workers are not mentioned in this framework and it may have a nonnegligible impact on their cultural acculturation and trust in physicians. In addition, a clear definition of acculturation is needed. Based on the results, it seems that adjusted for demographic covariates, socioeconomic covariates, and health profiles, only the LOS in Shanghai, good speaking skills of Shanghai dialect, and lifestyle integration are significantly associated with trust. I’m not sure if LOS in Shanghai and the good speaking skills of Shanghai dialect are good indicators for acculturation. Maybe a long time stay and learning local dialect do make people get more familiar with the healthcare system in Shanghai and have more experiences navigating the system, but these performances do not necessarily mean that he/she recognizes or integrates into the local culture.

4. The authors should also be aware of their ideological assumptions in the framework. For example, some research that used a critical theory framework may focus on structural inequality while others may concentrate on micro-level factors such as health literacy and focuses on psychosocial factors and cultural contexts. However, research focused on either side should focus on the concerns of the other side. For example, if we ignore the “oppressive structures” that migrant workers are facing, we may “lead to the pathologization of immigrants who may not be seen as acculturating ‘enough’, even if their behaviour is related to their experience of social exclusion or discrimination within the larger society”. (See Sakamoto, Izumi. "A critical examination of immigrant acculturation: Toward an anti-oppressive social work model with immigrant adults in a pluralistic society." British Journal of Social Work 37.3 (2007): 515-535.) Maybe the author could consider more variables about structural barriers to address this issue.

Method

Data collection

1. The legal permanent system or temporary residency permit system in China is quite complex. Maybe the author could add more description in the appendix for readers to understand. It would also provide a better rationale for the sampling strategy.

Measurement

2. The information about the questionnaire is inadequate, and the measurements of many variables are unclear. It would be useful to attach the questionnaire they used in the survey. For example, what questions the authors have asked for measuring “life integration”? And how this variable was dichotomized int to a binary variable (high vs low)?

Statistical analysis

3. Please state the rationale for the covariates that you have controlled. A DAG may be helpful (http://www.dagitty.net/dags.html).

Discussion & Conclusion

1. Hedonic recall bias: I believe the hedonic recall bias is more related to objective measurements, such as income. I wonder if the authors could show some evidence about the hedonic recall bias of psychosocial measurements.

2. The authors argued that their results “represent all of the top 4 largest cities of China (Beijing, Shanghai, Guangzhou, and Shenzhen)”. I can’t agree since these four tier-1 cities in China are very different in terms of geography, economics, and culture.

Minor concerns:

1. To my knowledge, the word “indigenous population” is used to differentiate the Indigenous peoples of the Americas from the European settlers. I guess here the authors are referring to the “rural population”. Also, the word “migrant worker” may be a better choice for “internal immigrant”.

2. The English language of the manuscript is not suitable for publication. For example, “the outstanding gap between the restricted supply of health service provided by the government and emergent medical demand from migrants” → “the large/huge gap between …” (Introduction).

3. Typos: “(p-value < 0.5)” → “(p-value < 0.05)” (Method). “10,000-250,000” → “100,000-250,000”. The mean (or median?) age for the “Good physician trust” and “Not trust physicians” groups are missing. The “Acculturation” is categorized as “low/high”, but the figures have five rows (Table 1).

4. “Gender” is referring to men and women. “Sex” is usually categorized as female or male.

5. Since this study is focusing on the association between acculturation and physician trust, the authors should avoid using words that suggest causality such as “effect”, “cause”, “affect”, and “impact”.

6. Several other variables may play important roles in this study such as the birthplace and the hukou of participants. I am not sure whether the authors have these measurements, but it would be interesting to see if the place of birth or hukou status is confounding factors or effect modifiers.

6. PLOS authors have the option to publish the peer review history of their article (what does this mean?). If published, this will include your full peer review and any attached files.

Reviewer #1: **Yes: **Jian Wang

Reviewer #2: No

---

## [Author Response · Author response to Decision Letter 0]

1 Oct 2022

Reviewers' comments:

Reviewer #1: I think this is a policy-relevant research article. I thought that the definition of internal migrants may varied across time. The authors may give a clear objective study population under the circumstance of a cross-section data time.

Response:Thank you for this suggestion. Accordingly,we have added the definition of study population:internal migrants into the section of introduction. please see highlighted lines in that section.

Reviewer #2: Thank you for considering me as a reviewer for this manuscript in your esteemed journal. In general, I believe that it’s an interesting study. The authors focus on an important research question. It also has a great potential for developing a practical, culturally safe tool that can be used to improve the patient experience of migrant workers and promote social cohesion in China. This study may also have implications for other countries that are experiencing the rise of urbanization. However, there are some key issues that I suppose the authors need to address before the manuscript can be considered for publication:

Major concerns:

Introduction

Motivation

1. I suggest our authors re-emphasize the importance of physician trust in health service research as well as its relationship with other important psychosocial measures that they have mentioned (e.g., physician-patient relationships, ratings of providers). For example, is there any evidence to show that physician trust has an impact on the health service utilisation or the health outcomes of patients? Is there any overlap between physician-patient relationships and physician trust?

Response :According to your suggestion, we have supplemented the statements on the association of physician trust with healthcare utilization, health outcomes of patients and physician-patient relationship. Please see highlighted lines in the section of introduction.

2. The authors mentioned that more than 60% of health care workplace violence in Shanghai was caused by migrant workers. It would have made the authors' research questions more meaningful if they had given the percentage of migrant workers in the population of Shanghai, or the percentage of migrant workers in outpatient/inpatient care utilisations.

Response：Thank for your recommendations, we have added the percentage of the internal migrants on outpatient and emergency department visits to demonstrate the research questions pointed out by the reviewer. Please see details highlighted in the section of introduction.

Conceptual framework

3. The authors should explain why “immigration acculturation theory” is suitable for studying migrant workers in China (since it was designed for immigrants in the Western world) and how they developed their framework (based on existing literature? Expert opinions? Or something else?). From my point of view, many structural barriers (e.g., hukou, health insurance, perceived discrimination) faced by migrant workers are not mentioned in this framework and it may have a nonnegligible impact on their cultural acculturation and trust in physicians. In addition, a clear definition of acculturation is needed. Based on the results, it seems that adjusted for demographic covariates, socioeconomic covariates, and health profiles, only the LOS in Shanghai, good speaking skills of Shanghai dialect, and lifestyle integration are significantly associated with trust. I’m not sure if LOS in Shanghai and the good speaking skills of Shanghai dialect are good indicators for acculturation. Maybe a long time stay and learning local dialect do make people get more familiar with the healthcare system in Shanghai and have more experiences navigating the system, but these performances do not necessarily mean that he/she recognizes or integrates into the local culture.

Response: Thanks for your detailed suggestions. Based on the prior literature (listed below as 1-3), immigration acculturation theory is also suitable for studying migrant workers or internal migrants in China due to the reason that internal migrants, as a part of large indigenous population, were observed to encounter difficulties of adapting into the cities caused by the existing intra-cultural differences between different regions and some structural barriers when they migrated from rural to urban areas. We added this description in the section of introduction (highlighted part). Meanwhile, we added the definition of acculturation according to your suggestion. Moreover, some previous studies have applied the indicators of acculturation with LOS dialect proficiency, dialect utilization and lifestyle adaption. Please see the highlighted lines in the sub-section of Measures.

[1]Miao S, Xiao Y. Does acculturation really matter for internal migrants’ health? Evidence from eight cities in China. Social Science & Medicine. 2020 Sep 1;260:113210.

[2]Fang L, Sun RC, Yuen M. Acculturation, economic stress, social relationships and school satisfaction among migrant children in urban China. Journal of Happiness Studies. 2016 Apr;17(2):507-31.

[3]Jihong Y.The Cultural Adaptation of New Urban Immigrants: A Case Study of Landless Farmers.Tianjin Social Sciences.2010;2:62-5.(Chinese)

4. The authors should also be aware of their ideological assumptions in the framework. For example, some research that used a critical theory framework may focus on structural inequality while others may concentrate on micro-level factors such as health literacy and focuses on psychosocial factors and cultural contexts. However, research focused on either side should focus on the concerns of the other side. For example, if we ignore the “oppressive structures” that migrant workers are facing, we may “lead to the pathologization of immigrants who may not be seen as acculturating ‘enough’, even if their behaviour is related to their experience of social exclusion or discrimination within the larger society”. (See Sakamoto, Izumi. "A critical examination of immigrant acculturation: Toward an anti-oppressive social work model with immigrant adults in a pluralistic society." British Journal of Social Work 37.3 (2007): 515-535.) Maybe the author could consider more variables about structural barriers to address this issue.

Response: Thank for your suggestion. In this study, we applied the acculturation theory and examined the association of acculturation with physician trust among internal migrants in China under the assumption that these structural barriers did not present effects on the trust in physicians. We agreed with the reviewer that the acculturation theory has some deficiencies, which largely ignore structural issues affecting individuals’ ability in dealing with unfamiliar culture(s). As the reviewer mentioned, the theory does not capture the big picture behind acculturating individuals and may lead to pathologization of immigrants who may not be seen as acculturating ‘enough’. Therefore, variables on structural barriers such as cultural distance, perceived discrimination, and hukou should be carefully considered to address the issues. But due to the survey methods we employed, it is hard for us to add these variables at this time point. We will certainly take into consideration of these important variables in our future research. We added the reviewer’s suggestion as a part of the research limitations into the revised manuscript. Please see the statements highlighted in the section of discussion.

Method

Data collection

1. The legal permanent system or temporary residency permit system in China is quite complex. Maybe the author could add more description in the appendix for readers to understand. It would also provide a better rationale for the sampling strategy.

Response:We have added a supporting information file to describe the legal permanent system or temporary residency permit system in China. Please see details in the S1 Table.

Measurement

2. The information about the questionnaire is inadequate, and the measurements of many variables are unclear. It would be useful to attach the questionnaire they used in the survey. For example, what questions the authors have asked for measuring “life integration”? And how this variable was dichotomized int to a binary variable (high vs low)?

Response: According to your recommendations, we displayed the survey questionnaire in the supporting information file(S2 File). The measurements of many variables can be seen in this file. In addition, we added the measurement approach of the variables of interest in the study. Please see highlighted lines in the sub-section of measures of the revised manuscript.

Statistical analysis

3. Please state the rationale for the covariates that you have controlled. A DAG may be helpful (http://www.dagitty.net/dags.html).

Response: According to your recommendation,we have supplemented the rationale for the covariates in the study with a DAG figure,please see details highlighted in the sub-section of statistical analysis and Fig 2.

Discussion & Conclusion

1. Hedonic recall bias: I believe the hedonic recall bias is more related to objective measurements, such as income. I wonder if the authors could show some evidence about the hedonic recall bias of psychosocial measurements.

Response: Thank for your correction. We have revised it into “recall bias”, not “hedonic recall bias”, which more often occurs in the self-reported objective measurements.

2. The authors argued that their results “represent all of the top 4 largest cities of China (Beijing, Shanghai, Guangzhou, and Shenzhen)”. I can’t agree since these four tier-1 cities in China are very different in terms of geography, economics, and culture.

Response: We have explained this in the section of Contributions and policy implications. Please see the details highlighted in that position.

Minor concerns:

1. To my knowledge, the word “indigenous population” is used to differentiate the Indigenous peoples of the Americas from the European settlers. I guess here the authors are referring to the “rural population”. Also, the word “migrant worker” may be a better choice for “internal immigrant”.

Response: In this study, the internal migrants refer to a population that has been living in a place other than their officially registered residence (hukou) for at least 6 months. This population does not only include the migrant workers flowing from rural areas to urban cities, but also the population who flow from one urban city into another different urban city. Therefore, we identify internal migrant as the study population instead of migrant worker you recommended.

2. The English language of the manuscript is not suitable for publication. For example, “the outstanding gap between the restricted supply of health service provided by the government and emergent medical demand from migrants” → “the large/huge gap between …” (Introduction).

Response: As you suggested, we have fully revised the language of this manuscript by a local English-speaking peer in the current version.

3. Typos: “(p-value < 0.5)” → “(p-value < 0.05)” (Method). “10,000-250,000” → “100,000-250,000”. The mean (or median?) age for the “Good physician trust” and “Not trust physicians” groups are missing. The “Acculturation” is categorized as “low/high”, but the figures have five rows (Table 1).

Response: Thanks for your correction, we have revised these accordingly. Please see highlighted lines where these typos occurred.

4. “Gender” is referring to men and women. “Sex” is usually categorized as female or male.

Response: We have revised gender into sex.

5. Since this study is focusing on the association between acculturation and physician trust, the authors should avoid using words that suggest causality such as “effect”, “cause”, “affect”, and “impact”.

Response: as you recommended, we replace the word “effect” with “assocation” , and revised the title of the manuscript accordingly.

6. Several other variables may play important roles in this study such as the birthplace and the hukou of participants. I am not sure whether the authors have these measurements, but it would be interesting to see if the place of birth or hukou status is confounding factors or effect modifiers.

Response: We agreed that the place of origin and hukou for immigrants are important variables in the immigration studies. However, in this study, we extracted 1330 participants as internal migrants who have been living in Shanghai instead of their officially registered residence (hukou) for at least 6 months, which means that their birthplace was not Shanghai and they did not have Shanghai’s hukou, so we didn’t include the two variables in the study. This criterion of eligible participant inclusion was mentioned in the sub-section of participants of Materials and Methods.

---

## [Decision Letter · Decision Letter 1]

3 Nov 2022

PONE-D-22-15202R1Association of acculturation with physician trust for internal migrants: a cross-sectional study in ChinaPLOS ONE

Dear Dr. Shi,

Thank you for submitting your manuscript to PLOS ONE. After careful consideration, we feel that it has merit but does not fully meet PLOS ONE’s publication criteria as it currently stands. Therefore, we invite you to submit a revised version of the manuscript that addresses the points raised during the review process. Additionally, please consider using a professional editing service or invite a native English speaker to polish the article. Otherwise, the current language is hard to follow and not appropriate for publication. 

We look forward to receiving your revised manuscript.

Kind regards,

Wenhui Mao, PhD

Academic Editor

PLOS ONE

Reviewers' comments:

Reviewer's Responses to Questions

**Comments to the Author**

1. If the authors have adequately addressed your comments raised in a previous round of review and you feel that this manuscript is now acceptable for publication, you may indicate that here to bypass the “Comments to the Author” section, enter your conflict of interest statement in the “Confidential to Editor” section, and submit your "Accept" recommendation.

Reviewer #2: (No Response)

2. Is the manuscript technically sound, and do the data support the conclusions?

Reviewer #2: Yes

3. Has the statistical analysis been performed appropriately and rigorously? 

Reviewer #2: Yes

4. Have the authors made all data underlying the findings in their manuscript fully available?

Reviewer #2: Yes

5. Is the manuscript presented in an intelligible fashion and written in standard English?

Reviewer #2: No

6. Review Comments to the Author

Reviewer #2: I'm happy to see the authors have addressed most of my comments and significantly imporved their manuscript. Here are my comments for your to consider:

1. In DAGs, time flows left to right. We usually draw exposures on the left and draw outcomes on the right. For example, if you think the number of chronic diseases of patients changed their trust to physicians, you should put it on the left side of physician trust. Since DAG is not the main methodology or result in your paper, you can put it in the appendices.

2. The authors still use the word "indigenous population" in the second paragraph of background. As I said in the previous round of review, the term indigenous first came into use by Europeans to describe Indigenous peoples of the Americas and this word is not suitable for internal migrants in China.

3. This manuscript still needs extensive revision for language and grammar. For example: "Association of acculturation with physician trust" → "Association between acculturation and physician trust", "salient association" → "significant association", "cut-point" → "cut-off point(or cut-off value)", "Physician Trust Good" → "High level of Physician Trust", "first-tier cities" → "Tier-1 cities". In addition, do not use the word like "liberal p value".

7. PLOS authors have the option to publish the peer review history of their article (what does this mean?). If published, this will include your full peer review and any attached files.

Reviewer #2: No

---

## [Author Response · Author response to Decision Letter 1]

3 Dec 2022

6. Review Comments to the Author

Reviewer #2: I'm happy to see the authors have addressed most of my comments and significantly imporved their manuscript. Here are my comments for your to consider:

1. In DAGs, time flows left to right. We usually draw exposures on the left and draw outcomes on the right. For example, if you think the number of chronic diseases of patients changed their trust to physicians, you should put it on the left side of physician trust. Since DAG is not the main methodology or result in your paper, you can put it in the appendices.

Response：Thank for your suggestion,we have adjusted the position of this variable, and put it in the appendix file S3.

2. The authors still use the word "indigenous population" in the second paragraph of background. As I said in the previous round of review, the term indigenous first came into use by Europeans to describe Indigenous peoples of the Americas and this word is not suitable for internal migrants in China.

Response:We have revised this statement for the word.

3. This manuscript still needs extensive revision for language and grammar. For example: "Association of acculturation with physician trust" → "Association between acculturation and physician trust", "salient association" → "significant association", "cut-point" → "cut-off point(or cut-off value)", "Physician Trust Good" → "High level of Physician Trust", "first-tier cities" → "Tier-1 cities". In addition, do not use the word like "liberal p value".

Response:We have a professional language editor review the manuscript before re-submission accordingly.

---

## [Decision Letter · Decision Letter 2]

8 Jan 2023

Association between acculturation and physician trust for internal migrants: a cross-sectional study in China

PONE-D-22-15202R2

Dear Dr. Shi,

We’re pleased to inform you that your manuscript has been judged scientifically suitable for publication and will be formally accepted for publication once it meets all outstanding technical requirements.

Kind regards,

Wenhui Mao, PhD

Academic Editor

PLOS ONE

Additional Editor Comments (optional):

Reviewers' comments:

Reviewer's Responses to Questions

**Comments to the Author**

1. If the authors have adequately addressed your comments raised in a previous round of review and you feel that this manuscript is now acceptable for publication, you may indicate that here to bypass the “Comments to the Author” section, enter your conflict of interest statement in the “Confidential to Editor” section, and submit your "Accept" recommendation.

Reviewer #2: All comments have been addressed

2. Is the manuscript technically sound, and do the data support the conclusions?

Reviewer #2: Yes

3. Has the statistical analysis been performed appropriately and rigorously? 

Reviewer #2: Yes

4. Have the authors made all data underlying the findings in their manuscript fully available?

Reviewer #2: Yes

5. Is the manuscript presented in an intelligible fashion and written in standard English?

Reviewer #2: Yes

6. Review Comments to the Author

Reviewer #2: I think that the authors have adequately addressed the comments made by the reviewers in the

revised version of the manuscript. Therefore, I have no further comments.

7. PLOS authors have the option to publish the peer review history of their article (what does this mean?). If published, this will include your full peer review and any attached files.

Reviewer #2: No

---

## [Editor Report · Acceptance letter]

27 Feb 2023

PONE-D-22-15202R2 

Association between acculturation and physician trust for internal migrants: a cross-sectional study in China 

Dear Dr. Shi:

I'm pleased to inform you that your manuscript has been deemed suitable for publication in PLOS ONE. Congratulations! Your manuscript is now with our production department. 

Kind regards, 

on behalf of

Dr. Wenhui Mao 

Academic Editor

PLOS ONE